# Quantifying Downstream Climate Impacts of Sea Surface Temperature Patterns in the Eastern Tropical Pacific Using Clustering

Jason Finley [1,*], Boniface Fosu [1], Chris Fuhrmann [2], Andrew Mercer [1] and Johna Rudzin [1]

[1] Department of Geosciences, Mississippi State University, Starkville, MS 39762, USA; bof20@msstate.edu (B.F.); mercer@gri.msstate.edu (A.M.); johna.rudzin@geosci.msstate.edu (J.R.)
[2] Department of Geography and Environment, University of North Carolina, Chapel Hill, NC 27599, USA; fuhrman1@unc.edu
[*] Correspondence: jpf223@msstate.edu

**Abstract:** El Niño–Southern Oscillation (ENSO) phases and flavors, as well as off-equatorial climate modes, strongly influence sea surface temperature (SST) patterns in the eastern tropical Pacific and downstream climate. Prior studies rely on EOFs (which characterize fractional SST variance) to diagnose climate-scale SST structures, limiting the ability to link individual ENSO flavors with downstream phenomena. Hierarchical and k-means clustering methods are used to construct Eastern Pacific patterns from the ERSST dataset spanning 1950 to 2021. Cluster analysis allows for the direct linkage of individual SST years/seasons to ENSO phase, providing insight into ENSO flavors and associated downstream impacts. In this study, four clusters are revealed, each depicting unique SST patterns influenced by ENSO and Pacific Meridional Mode (PMM) phases. A case study demonstrating the utility of the clusters was also carried out using accumulated cyclone energy (ACE) in the Atlantic and Eastern Pacific basins. Results showed that Eastern Pacific (EP) El Niño suppresses Atlantic tropical cyclone (TC) activity, while Central Pacific (CP) La Niña enhances it. Further, EP El Niño, coupled with positive PMM, amplifies ACE. Ultimately, the methods used herein offer a cleaner analysis tool for identifying dominant SSTA patterns and employing those patterns to diagnose downstream climatic effects.

**Keywords:** clustering; unsupervised machine learning; ENSO; PMM; sea surface temperatures; eastern tropical Pacific; tropical cyclones

## 1. Introduction

The El Niño–Southern Oscillation (ENSO), a coupled atmospheric–oceanic phenomenon in the tropical Pacific, is the dominant source of interannual sea surface temperature (SST) and global climate variability [1–3]. ENSO produces anomalous atmospheric circulation patterns that can cause extreme weather events that impact human life and property [4–9]. Though extensive research has established ENSO influences on large-scale climate (i.e., teleconnections) that help forecast global temperature and precipitation patterns, and more recent work has discovered diversity among types of ENSO events (so-called ENSO "flavors"), the impacts of these ENSO types on downstream weather and climate in regions such as North America is less established [6,8].

ENSO flavors largely differ by the location of SST anomalies (SSTAs) within the equatorial Pacific and the patterns of SSTA evolution [6,8]. For example, the two most common El Niño types are the Eastern Pacific type (EP), also known as the canonical El Niño, and the Central Pacific type (CP), which is often referred to as CP El Niño or El Niño Modoki [8,10–12]. For canonical events, peak SSTAs emerge in the eastern equatorial Pacific during boreal spring and progress toward the Central Pacific in boreal fall. In contrast, Modoki events are linked to large SSTAs near the dateline, evolve from anomalies in the subtropical North Pacific, and begin to

emerge during boreal spring or summer [8]. The El Niño Modoki is a more recent discovery that produces global impacts that differ from the effects of canonical El Niño events [10,13]. Given that the tropospheric temperature response lags ENSO-related SSTA changes in the order of a few months [14], the timing of ENSO impacts may depend on ENSO flavor due to shifted locations of maximum SSTAs [15]. The diverse impacts on wintertime atmospheric circulation patterns from EP and CP ENSO events are due, in part, to the rising branch of the Walker Circulation generally being found above the warmest waters in the equatorial Pacific [11,16–18]. Furthermore, convection over the warmest waters leads to the release of latent heat, which in turn warms the middle and upper troposphere within the tropics. This warmer-than-usual air then moves towards the poles as part of the Hadley Circulation [19]. However, this warmer air results in a stronger meridional temperature gradient aloft between the tropics and midlatitudes, which in turn strengthens the jet stream. This often results in the jet stream shifting farther south than usual during the boreal winter [20]. This shifted jet stream steers extratropical cyclones toward the southern tier of the U.S., leading to increased precipitation in this region. Meanwhile, the northern U.S. and parts of southern Canada experience relatively dry conditions [7].

Prior research has explored the utility of Empirical Orthogonal Functions (EOFs) [21] in characterizing Central Pacific SST patterns. For example, Ashok et al. [11] employed EOF analysis to demonstrate that El Niño Modoki events are represented by the second mode, explaining 12% of the variance in monthly tropical Pacific SST anomalies. Similarly, Takahashi et al. [22] suggested that the first two EOF modes of tropical Pacific SST anomalies describe nonlinear flavors of ENSO. Additionally, Zhao et al. [23] and Vimont et al. [24] emphasized the role of EOF in classifying ENSO pattern diversity and distinguishing between different types or flavors of ENSO events based on SST anomalies.

While these studies underscore the utility of EOFs in characterizing variability modes within the SSTs, individual EOFs describe a percentage of the total variability of SST [21]. Statistical corrections are required to utilize EOFs to make representative composite maps, and these maps rarely effectively characterize an observed seasonal SST pattern, since each composite pattern comprises a fraction of the total SST variability. This issue is further exacerbated when trying to link individual EOFs to observed SST patterns on varying (seasonal and annual) time scales, since each year/seasonal SST pattern comprises a linear combination of EOFs. However, such linkages are critical to understanding downstream impacts effectively. This important limitation is a primary motivation for our study.

Prior EOF methods have revealed additional modes of climate variability in North Pacific SSTs (those with meridional and off-equatorial components) that interact with ENSO (such as the Pacific Meridional Mode (PMM)), convoluting the characterization of downstream impacts [25,26]. The PMM consists of a north–south SST gradient related to trade wind anomalies in the subtropical North Pacific [27,28]. The fluctuating zonal distribution of SSTs and atmospheric responses across ENSO phases influence convection within the tropics as well as temperature and precipitation patterns over North America through modification of the off-equatorial SST patterns [25,26,29]. Stuecker [30] utilized an EOF methodology to investigate CP ENSO events influenced by the PMM. Fan et al. [31] and Kao et al. [32] highlighted the importance of understanding the impact of the PMM on the North Atlantic and North Pacific using EOF. Messie and Chavez [33] conducted EOF analysis over 100 years of global SSTs, resulting in two modes identified as ENSO and the Pacific Decadal Oscillation (PDO).

While EOFs have shown clear benefits in identifying important structures underlying Pacific SST variability, the resulting patterns are still limited to explaining a fraction of an actual timestep's variability. Cluster analysis offers several benefits when compared with EOFs for identifying and understanding sea surface temperature (SST) patterns and related impacts on climate [6]. While EOF analysis is proficient in identifying dominant modes of variability, cluster analysis helps pinpoint coherent spatial patterns and individual events associated with each pattern, and clusters are directly constructed using observed SST fields, not fractions of the variability of the SST [21]. This allows for the identification

of distinct SST regimes and their associated impacts [34]. Further, EOFs are inherently linearly orthogonal, resulting in a decomposition of variance that assumes linear relationships among the structures. Cluster analysis makes no such assumption, allowing for the inclusion of nonlinear relationships when constructing composite SST fields [35]. As such, unsupervised machine learning methods such as cluster analysis treat the atmosphere as a continuum without enforcing linearity assumptions and can produce patterns that better align with actual observations when compared with linear techniques such as EOFs [36].

As an example, Schulte et al. [34] found a strong correlation between the EOF-based East Pacific–North Pacific (EP-NP) pattern and water temperatures near the Long Island Sound. However, their result was based on rotated principal component analysis (a highly similar method to EOFs that reveals fractional variance explained per EOF), which limited their ability to diagnose the specific role of the EP-NP pattern for a given December. That is, while the method demonstrated overall underlying variability modes, it could not be readily applied to compare against a specific observational example, which limited its utility in such an application. More recent work has explored the use of cluster analysis when exploring ENSO and PMM SST characteristics. Su et al. [37] used cluster analysis to better understand the pendulum of ENSO phases, and Zhao et al. [38] utilized clustering to find the diversity of the PMM in the North Pacific. These studies highlight the usefulness of clustering but do not focus on diagnosing downstream impacts.

According to Stuecker [30] and Vimont [28], subsequent Pacific SST work should focus on both the PMM and ENSO since they are intrinsically linked at interannual timescales. While the relationships among the PMM and ENSO are somewhat well defined, SST patterns that characterize the underlying structures are limited to those patterns derived from EOFs with the associated limitations discussed above. To date, no study has removed the assumption of linearity and developed patterns that directly comprise constituent events (i.e., years) that can be used to diagnose downstream impacts of these ENSO flavors. As such, the primary objective of this paper is to apply hierarchical and k-means clustering to identify established off-equatorial and ENSO-related SSTA patterns in the Eastern Pacific from 1950 to 2021 and link those clusters back to their constituent member years. These patterns, which strongly align with well-understood ENSO flavors (see below), will have much greater utility in downstream impact application studies in future work. Given the recent surge in the popularity of unsupervised learning methods not previously applied in this research context, such as cluster analysis, employing these methods will reinforce existing prevalent SST patterns with the flexibility of linking those patterns directly to constituent years, which is a critical outcome of this work. To demonstrate this utility, we offer an example application for North America using accumulated cyclone energy (ACE) by tropical cyclone (TC) season to identify statistically significant differences in ACE by derived SST pattern/ENSO flavor.

Section 2 will address materials and methods, while Section 3 will provide results. We summarize the results in Section 4.

## 2. Materials and Methods

### 2.1. Data

This study explores data from the 2° × 2° NOAA Extended Reconstructed Sea Surface Temperature (ERSST) Version 5 dataset, available monthly, beginning in January 1854 [39]. Due to large spatial gaps in SST data during the 1800s and early 1900s as well as changes in measuring SSTs in the 1940s, the most reliable SST data span from 1950 to present [39]. To reduce the effects of the long-term SST trend from 1950 to 2021, this study computes SSTAs using 30-year centered means following the Climate Prediction Center's (CPC) latest approach for calculating the Oceanic Niño Index (ONI) [40–42]. The ONI, computed as the 3-month average of SSTAs in the Niño 3.4 region (5° S–5° N, 120–170° W), supports ENSO classification later in this study. The 30-year centered means are centered on 5-year periods and computed over a span starting 14 years prior and 11 years after a given 5-year period [42]. For example, during 1950–1954, the 30-year base period is 1936 (14 years before

1950) to 1965 (11 years after 1954). Then, from 1955 to 1959, the 30-year base period is 1941 to 1970, and so on. This technique enables improved classifications of ENSO episodes in the earlier part of the period (i.e., the 1950s) by eliminating the use of more recent, higher SST averages for assessing older ENSO events.

Note that although this approach incorporates SST data before 1950, Huang et al. [39] improved data quality for the 1930s and 1940s through spatial filtering and cross-validation with independent data sources, an update that has been referenced in the literature (e.g., [41,43]). Furthermore, an analysis of variance during 1936–2021 at each grid point within the study area utilizing 30-year periods as the means indicated only 1 to 2% contribution to variance from any changes in the dataset, instead depicting the most variable regions being linked to ENSO and the PMM.

The study domain spans 30° N to 20° S and 180° W to 80° W, encompassing an area beyond off-equatorial SSTs (10° N to 20° N) in the North Pacific to include the equatorial Pacific and off-equatorial South Pacific, following the latitudinal extent used to identify the PMM [27]. In this study, SSTAs are evaluated over three-month periods associated with boreal spring (MAM), summer (JJA), fall (SON), and winter (DJF). These periods facilitate analysis of the seasonal evolution of SSTs (e.g., the PMM, which typically peaks in boreal spring, and ENSO, which is usually dominant in the boreal winter).

Indices based on SST data are also used in this study. The ONI, as defined above, is calculated using the three-month average of SSTs in the tropical Pacific between 120° and 170° W [41]. The phase and strength of an ENSO event are measured using this metric. The Pacific Meridional Mode (PMM) index is computed using the approach developed by Chiang and Vimont [27]. This involves applying a maximum covariance analysis (MCA) [44] to SSTs and 10 m winds from 1950 to 2005 between 175° E and 95° W and between 21° N and 32° N, which is the area that the PMM covers. MCA uses a singular value decomposition of the covariance matrix of the SST and wind fields, with seasonal and linear trends removed [45]. Additionally, the Pacific Cold Tongue index is removed to ensure that the PMM index has limited covariability with ENSO. The final PMM index used in this study is derived by projecting the MCA-derived PMM spatial pattern onto the SST values [27].

Since the cluster analysis method discussed above can link SST seasons to given underlying SST patterns, we included 10 m winds to establish near-surface momentum structure for the SST seasons. Wind data were obtained from the European Center for Medium-Range Weather Forecasts (ECMWF) ERA5 Reanalysis [46]. The ERA5 data are provided on a 0.25° × 0.25° grid (approximately 31 km) using 137 levels from the surface up to about 0.01 hPa (about 80 km). These data are based on hourly estimates for a wide range of climate variables, including atmospheric, land, and ocean, starting in 1940, though we retained 10 m winds exclusively for this study.

Finally, to provide an example application of the SST analysis to a downstream research problem, we obtained accumulated cyclone energy (ACE) for the Eastern Pacific and Atlantic basins. In ACE studies, the Atlantic basin stretches eastward to the Cape Verde islands, encompassing the Gulf of Mexico and the Caribbean, while the Eastern Pacific basin covers all Pacific locations north of the equator and east of the international dateline [47]. ACE is a widely accepted metric used to quantify the overall activity and intensity of TCs within a specific basin over a given time period [48]. ACE is based on the HURDAT2 data [49] and calculated by summing the squares of the maximum sustained wind speeds of TCs at 6 h intervals throughout their lifetimes [47]. ACE is calculated for all named storms (tropical storm strength or higher) within a particular basin [47,48]. This index takes into account both the duration and intensity of each system [47,48]. This metric provides an accurate representation of the total energy released by TCs rather than simply counting the number of storms or their wind speeds [50] and is thus a useful TC measure for climatological analysis.

### 2.2. Cluster Analysis

To characterize SSTA patterns in the study area, hierarchical and *k*-means cluster analyses are used [21]. Cluster analysis is an unsupervised learning method that groups data (clusters) based on statistical similarities and differences within that dataset [21]. A distance metric is used to cluster data—here, the Euclidean distance:

$$d_{ij} = ((x_i - x_j)(x_i - x_j)^T)^{1/2}, \tag{1}$$

where $x_i$ and $x_j$ in our study represent individual SST patterns in our study region. Cluster analysis groups patterns based on their statistical proximity, such that those events grouped together have similar data structures. An advantage (and challenge) with cluster analysis is the lack of a priori knowledge of groups/clusters, which can reveal previously unseen patterns in some instances when compared with other approaches [21]. Hierarchical cluster analysis groups events (here, SST patterns) by identifying elements whose statistical distance is smallest and pairing them. The method then repeats until all points are paired or begins linking up groups of these pairs using a linkage method. This method is then built into a cluster hierarchy that shows how the underlying events group together. Visually, this hierarchy is known as a dendrogram (Figure 1). *K*-means clustering uses a similar approach, but instead utilizes Euclidean distance relative to a randomly generated cluster center (based on a first guess number of clusters). From here, cluster centers adjusted as points are added to each cluster until the solution converges on a final set of cluster centers (no hierarchy is created).

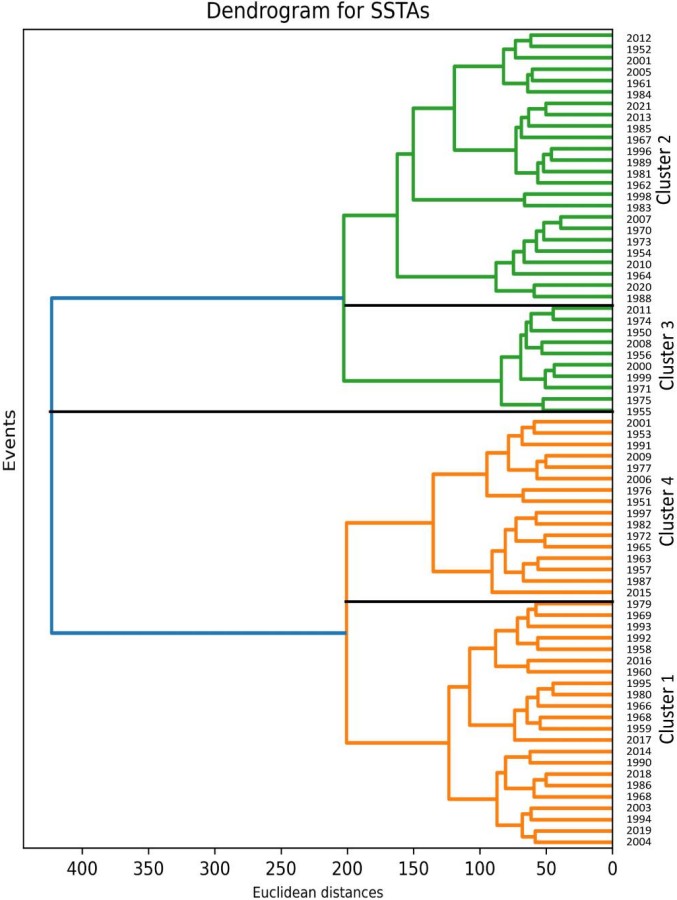

**Figure 1.** Dendrogram from hierarchical clustering used to assess the number of clusters. The black lines within the dendrogram separate and highlight the four clusters. Note that the cluster numbers in this dendrogram may not match the cluster numbers from *k*-means clustering.

For our study, we used a blended hierarchical and k-means clustering methodology. First, we utilize hierarchical clustering (Figure 1) to provide a first guess of the correct number of clusters to retain for the full SST database (e.g., all seasons). We then apply k-means cluster analysis across both time and spatial dimensions to ensure the coherency of the resulting patterns in both time and space. That is, regardless of season, a given cluster will comprise the same subset of SST years.

As stated previously, k-means clustering requires a first guess of the number of clusters. Since centers are generated randomly, we repeated the *k*-means procedure 25 times to ensure that the results were not sensitive to the random initialization. However, the resulting clusters were not sensitive to the random initialization (the algorithm converged consistently on the same solution). *K*-means clustering is repeated for different numbers of clusters [21,51] because the underlying cluster structure is often not known.

The silhouette coefficient (Equation (2); [21]), which measures both the intra-cluster spread and inter-cluster spread, is used to evaluate the quality of the resulting clusters:

$$S(i) = \frac{b(i)\text{-}a(i)}{\max\{a(i), b(i)\}},$$ (2)

where b(i) represents the smallest average distance of a point (SST field in our study) to points in other clusters, and a(i) represents the average distance of a point from the other points in the cluster or group to which it belongs [21]. If the average of (2) is close to 1, then most, if not all, points are correctly assigned, and the clusters are well separated [51]. If the average coefficient is close to 0, then clusters may overlap, and some points may be between clusters. If the value is negative, then many points are likely incorrectly assigned [21,51].

The silhouette metric S* (Equation (3)) captures the global average of silhouette coefficients and the number of misclustered events (those events with a negative silhouette coefficient) for each period [52].

$$S* = \left(\frac{N\text{ - }m_s}{N}\right)\overline{S},$$ (3)

where $\overline{S}$ is the average silhouette coefficient for all members (the average of S(i)), $m_s$ is the number of misclustered members, and N is the sample size. This metric modifies the average silhouette coefficient ($\overline{S}$) by using the percentage of negative S values. If $m_s$ is 0, then S* is the same as $\overline{S}$. In addition to S*, the Pearson correlation is calculated between the composite map for each cluster and each member of the cluster [52]. Each of these correlations is then averaged to assess the degree of representativeness of the constituent members for each cluster. This correlation also helps to identify the ideal cluster number, where the weighted mean is used to give less weight to clusters that have a small number of cases (and more weight to the clusters with a larger number of cases). Multiple numbers of clusters (two to nine clusters) are assessed to determine the optimal cluster number. In this study, using two clusters resulted in one cluster simply representing the mean of SSTs for the associated three-month period, while nine or more resulted in clusters with only one or two samples. The optimal cluster number for each period is found to be four by exploring the dendrograms, weighted mean S*, and Pearson correlation. The four groups of clusters with N = 24, 22, 12, and 14 members have a weighted mean Pearson correlation of 0.46 and a weighted mean S* of 0.12 with no misclustered members.

Once the clusters were identified and cases were sorted into each group, the members of each cluster/group were averaged at each grid point, yielding seasonal composite SST fields for each cluster from which we could begin exploring the resulting SST patterns and downstream effects. To better understand the interannual variability of each cluster, a time series was created for each cluster. We kept only 1950 to 2019 to cleanly analyze decadal frequency. To demonstrate the utility of these patterns in describing downstream effects, we also conducted a simple application study exploring how ACE relates to each

SST cluster. We used 10,000-iteration bootstrap mean ACE values for each year to establish the statistical significance of ACE across clusters.

## 3. Results

### 3.1. Resulting SST Cluster Patterns

A composite of seasonal SSTA patterns per cluster, along with each cluster's decadal frequency, is depicted in Figures 2–5. Cluster 1 (Figure 2) is characterized by a horseshoe pattern of positive anomalies in the subtropical North Pacific, Central Pacific, and central and eastern South Pacific coupled with weak anomalies in the Eastern Pacific. This pattern closely resembles a positive PMM, which typically consists of positive SSTAs that extend from Baja California to the central equatorial Pacific and into the subtropical South Pacific [27]. This pattern is most prominent during MAM, which commonly evolves into a CP El Niño event by DJF. The seasonal evolution of this cluster highlights the PMM during MAM as a precursor to CP El Niño conditions during DJF in a subset of cases (9 out 24 cases) [53]. Wind anomalies over the subtropical and equatorial Pacific are predominately westerly (albeit weak), indicative of the Wind–Evaporation–SST (WES) feedback that affects low-level winds and subsequently facilitates the migration of warm water in the subtropical North Pacific to the Central Pacific [27]. The decadal frequency of years (Figure 2E) within cluster 1 increased over the time period. The overall average increase in frequency corresponds to an increase in positive PMM events. Approximately 78% of cluster 1 member years had a positive PMM.

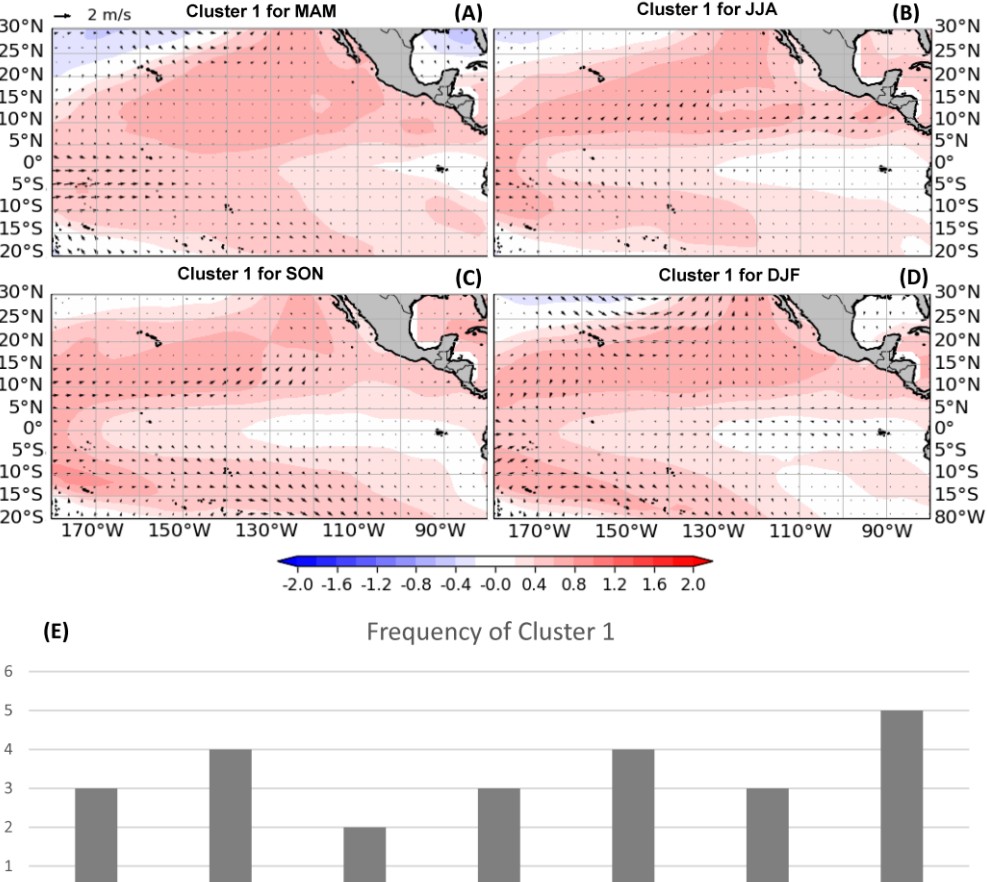

**Figure 2.** Composite patterns of standard anomalies of SST for cluster 1 for (**A**) March–April–May (MAM); (**B**) June–July–August (JJA); (**C**) September–October–November (SON); (**D**) December–January–February (DJF). The arrows represent wind anomalies in ms$^{-1}$. Decadal frequency of cluster 1 (**E**) from 1950 to 2019.

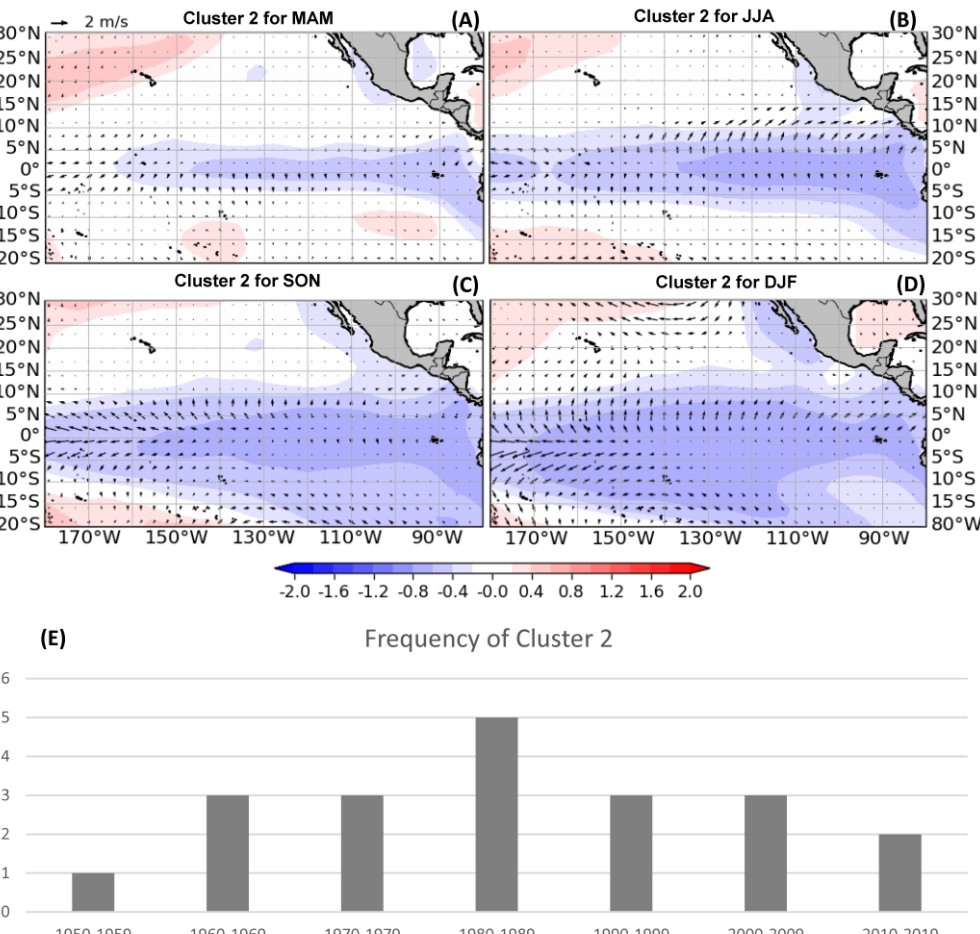

**Figure 3.** Composite patterns of standard anomalies of SST for cluster 2 for (**A**) March–April–May (MAM); (**B**) June–July–August (JJA); (**C**) September–October–November (SON); (**D**) December–January–February (DJF). The arrows represent wind anomalies in ms$^{-1}$. Decadal frequency of cluster 1 (**E**) from 1950 to 2019.

Cluster 2 (Figure 3) is dominated by weakly negative anomalies across the equatorial Eastern and Central Pacific, accompanied by small pockets of negative values off the coast of Mexico. This pattern signifies an EP La Niña event, which evolves from an intensification of negative anomalies primarily situated along the equator during the spring to a more expansive area of negative anomalies extending from the equatorial region to the subtropical Southern Hemisphere [53]. Weak wind anomalies in the subtropical North Pacific and easterly wind anomalies along the equator further corroborate EP La Niña conditions. The frequency of cluster 2 has seen a minor increase over the time period (Figure 3E). However, when you exclude the decade from 1980 to 1989, which has an unusually high number of cases, there is no pronounced decadal trend.

Cluster 3 (Figure 4) is characterized by below-normal values spanning most of the Eastern and Central Pacific. The most pronounced anomalies are observed in the Central Pacific near the equator and in subtropical areas (approximately between 15° N and 10° S) east of the dateline (approximately between 130° W and 180° W). This pattern indicates a negative PMM event during all periods, which can evolve into a robust CP La Niña in the subsequent winter [25,53]. The seasonal evolution in this cluster shows that a negative PMM during MAM often precedes a CP La Niña during DJF in 11 out of the 12 cases. The patterns and evolution of negative PMM events are similar to positive PMM events but with negative SSTAs [27]. Wind anomalies are northeasterly in the subtropical regions of the North Pacific during MAM, which helps force cooler waters from the subtropical North Pacific to move towards the central equatorial Pacific. Subsequently, strong easterly

wind anomalies develop near the equator in the Central Pacific during DJF, reinforcing a CP La Niña event [53]. Cluster 3 exhibits a slight decrease across the period (Figure 4E). The occurrence of negative PMM events has been decreasing during this period [54,55], implying that this decrease has an impact on the decline of the frequency of cluster 3.

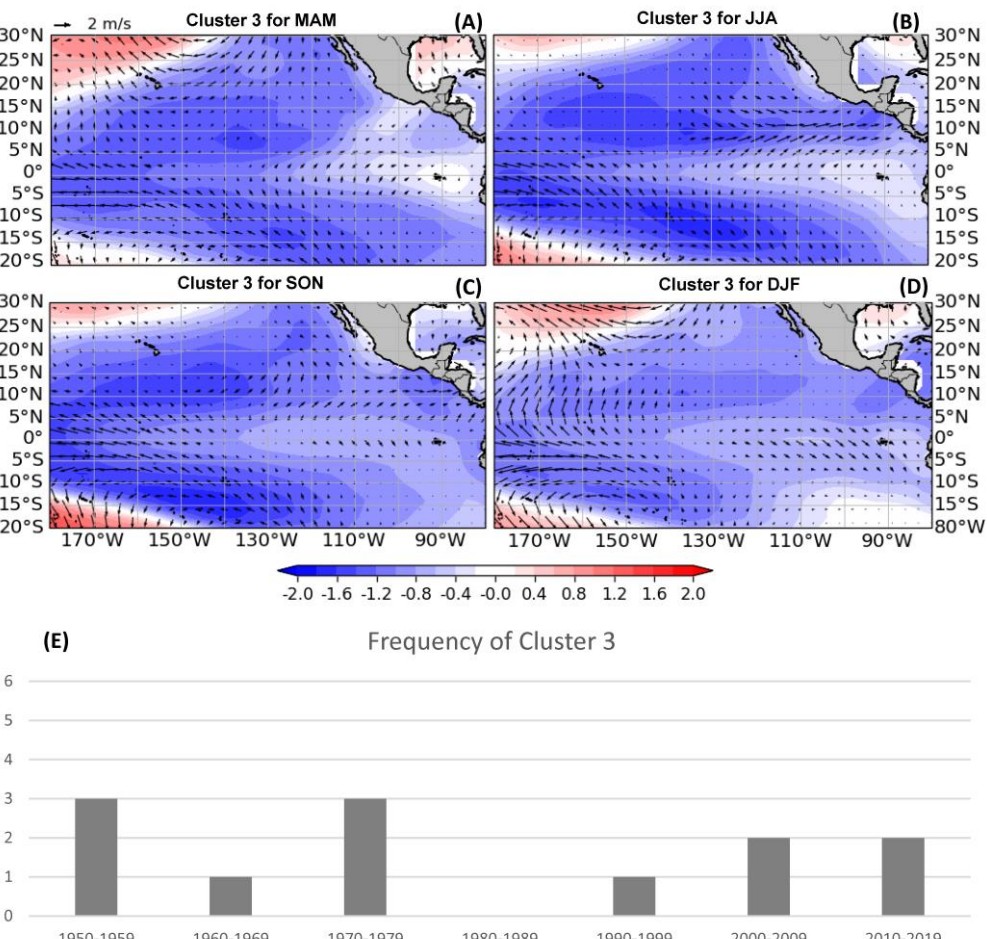

**Figure 4.** Composite patterns of standard anomalies of SST for cluster 3 for (**A**) March–April–May (MAM); (**B**) June–July–August (JJA); (**C**) September–October–November (SON); (**D**) December–January–February (DJF). The arrows represent wind anomalies in ms$^{-1}$. Decadal frequency of cluster 1 (**E**) from 1950 to 2019.

Cluster 4 (Figure 5) is dominated by positive anomalies across much of the equatorial Eastern Pacific, with the largest anomalies between 10° N and 10° S. Predominantly weakly positive anomalies are evident across the equator during MAM. Similarly, weak anomalies are noticeable in the subtropical North Pacific during JJA and SON. However, by DJF, a more robust EP El Niño emerges. A strongly positive PMM signature is absent during the spring, which further substantiates an EP El Niño [53]. Wind anomalies in the subtropical North Pacific are predominately easterly during JJA and SON. However, particularly during DJF, the wind anomalies are westerly over the equatorial Pacific. The frequency of cluster 4 (Figure 5E) has decreased slightly. The decreasing frequency may be due to the negative PMM, since 73% of events in cluster 4 are associated with a negative PMM. This is despite the prevalence of an El Niño signature in this cluster, and the overall number of El Niño events is increasing based on the ONI.

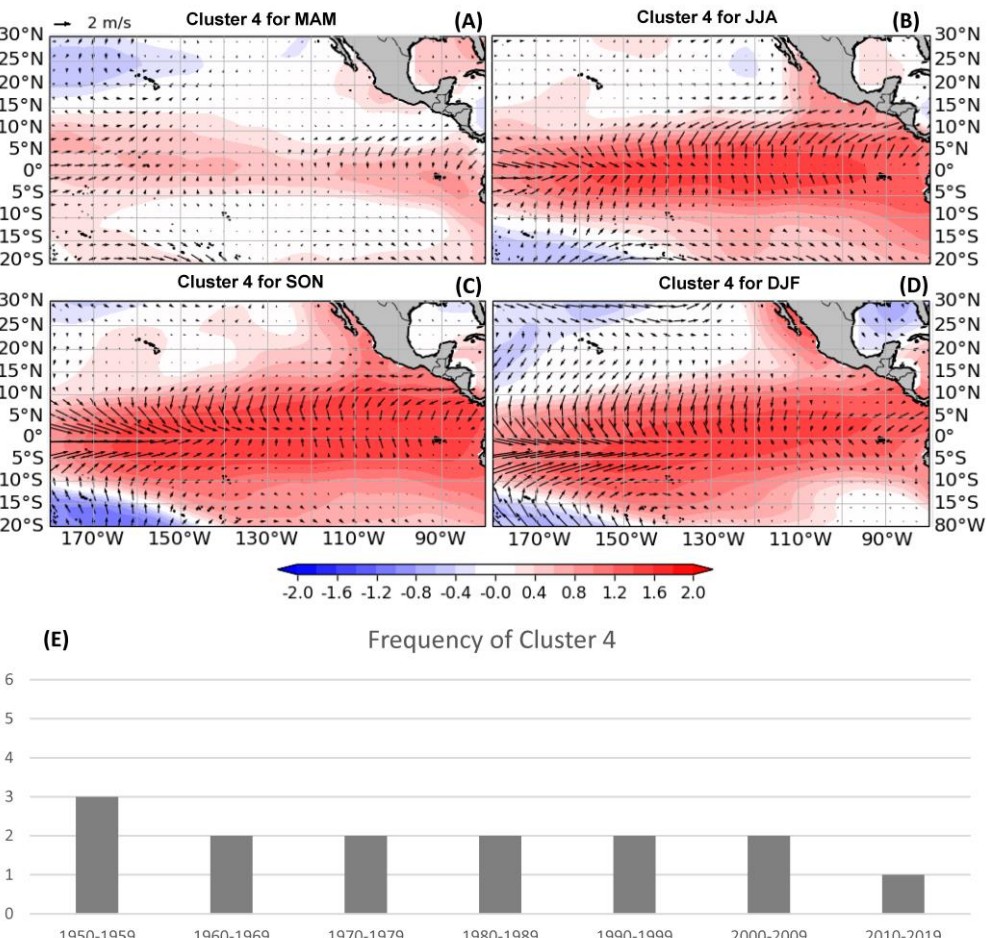

**Figure 5.** Composite patterns of standard anomalies of SST for cluster 4 for (**A**) March–April–May (MAM); (**B**) June–July–August (JJA); (**C**) September–October–November (SON); (**D**) December–January–February (DJF). The arrows represent wind anomalies in ms$^{-1}$. Decadal frequency of cluster 1 (**E**) from 1950 to 2019.

### 3.2. Analysis of the Oceanic Niño Index (ONI) and PMM

Figure 6 represents a scatterplot of all members of each cluster, with ONI values along the *x*-axis and PMM index values along the *y*-axis. Table 1 includes the number of ENSO and PMM events broken down into phases (i.e., La Niña, neutral, El Niño, negative PMM, and positive PMM). This analysis is a direct result of having the years/members of the clusters and further supports the SSTA patterns identified within each cluster. The majority of the cluster members in Figure 6 are relatively well separated, especially during fall and winter, where cluster 1 (4) has positive ONI and positive (negative) PMM values, and cluster 2 (3) has negative ONI and positive (negative) PMM values.

In cluster 1, the majority of members are characterized by positive ONI and PMM values, especially during summer (Figure 6B), fall (Figure 6C), and winter (Figure 6D). These results are consistent with the dominant neutral ENSO conditions and positive PMM events found in Table 1, with a notable shift towards El Niño in the winter. The average ONI for cluster 1 ranges from 0 to 0.5, indicative of neutral ENSO events with a tendency toward weak El Niño conditions [56]. The mean PMM values are strongly positive, ranging from 1.59 during spring to 1.93 during fall and winter. Cluster 2 predominantly exhibits negative ONI but positive PMM values during fall (Figure 6C) and winter (Figure 6D). Table 1 shows a high proportion of positive PMM events and neutral ENSO conditions, with La Niña events becoming more frequent in the winter. The mean ONI for cluster 2 for each period falls between 0 and −0.5, implying neutral ENSO conditions with a tendency toward weak La Niña conditions [56]. Mean PMM values are negative for MAM and

weakly positive for all other periods, which suggests this cluster is also influenced by weak PMM events.

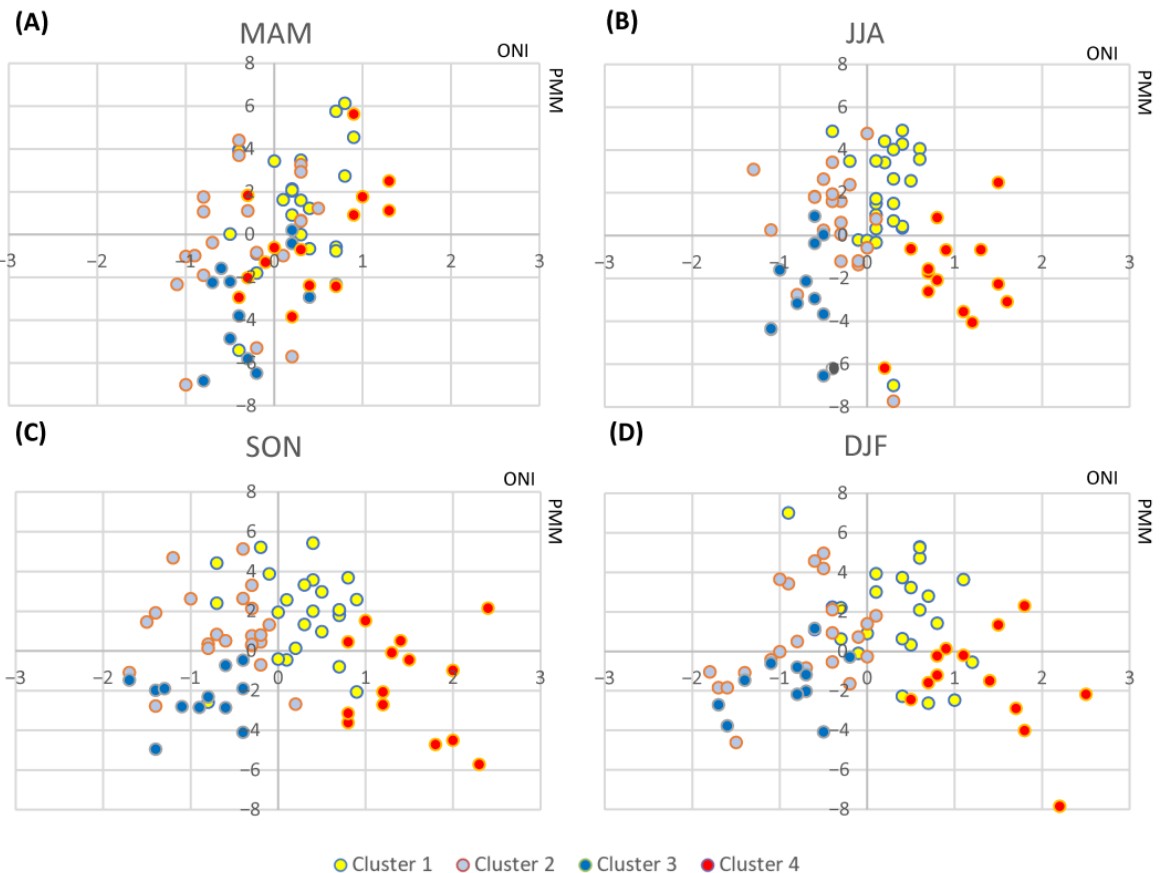

**Figure 6.** Scatterplot of all members of each cluster with ONI values along the *x*-axis and PMM index values along the *y*-axis for (**A**) MAM; (**B**) JJA; (**C**) SON; and (**D**) DJF.

For cluster 3, both ONI and PMM values are mostly negative for all seasons (Figure 6). This corresponds with the majority of events classified as La Niña and negative PMM in Table 1. The mean ONI is negative for all periods, indicating strong La Niña conditions, with the largest values during fall and winter. The mean PMM index is negative for all seasons, with spring having the most negative mean value of −3.28. This result is consistent with the peak of the PMM generally occurring during boreal spring [21]. For cluster 4, most ONI values are positive but the PMM values are mixed in the spring (Figure 6A). However, during summer, fall, and winter, the majority of PMM values are negative (Figure 6B–D). All periods except spring have mean ONI values above 0.5, indicating a robust El Niño. These results are supported by Table 1, which shows a majority of events characterized by El Niño and negative PMM conditions.

**Table 1.** The number of ENSO events (La Niña, neutral, El Niño) and PMM events for each season and cluster. ENSO and PMM phases are based on the ONI and PMM index, respectively.

| Cluster Number | N | Season | La Niña | Neutral | El Niño | Negative PMM | Positive PMM |
|---|---|---|---|---|---|---|---|
| 1 | 24 | MAM | 1 | 17 | 6 | 7 | 17 |
| | | JJA | 0 | 21 | 3 | 4 | 20 |
| | | SON | 3 | 12 | 9 | 5 | 19 |

**Table 1.** *Cont.*

| Cluster Number | N | Season | La Niña | Neutral | El Niño | Negative PMM | Positive PMM |
|---|---|---|---|---|---|---|---|
| | | DJF | 2 | 10 | 12 | 5 | 19 |
| 2 | 22 | MAM | 9 | 12 | 1 | 11 | 11 |
| | | JJA | 7 | 15 | 0 | 7 | 15 |
| | | SON | 10 | 12 | 0 | 4 | 18 |
| | | DJF | 14 | 8 | 0 | 11 | 11 |
| 3 | 12 | MAM | 5 | 6 | 1 | 11 | 1 |
| | | JJA | 11 | 1 | 0 | 10 | 2 |
| | | SON | 9 | 3 | 0 | 12 | 0 |
| | | DJF | 11 | 1 | 0 | 11 | 1 |
| 4 | 14 | MAM | 0 | 8 | 6 | 8 | 6 |
| | | JJA | 0 | 1 | 13 | 12 | 2 |
| | | SON | 0 | 0 | 14 | 10 | 4 |
| | | DJF | 0 | 0 | 14 | 11 | 3 |

### 3.3. Accumulated Cyclone Energy (ACE) Case Study

This study also examines ACE to demonstrate the effectiveness of cluster analysis in evaluating downstream climatic impacts for North America. ACE has been utilized to study the relationship between TC activity and climate phenomena, such as El Niño and La Niña events, providing insights into the influence of large-scale climate patterns on cyclone intensity [57]. Importantly, in our study, ACE can be directly linked to the SST patterns via the cluster member years, allowing for a cleaner investigation of the behavior of ACE in both the Atlantic (Figure 7) and Eastern Pacific (Figure 8) basins across our study period.

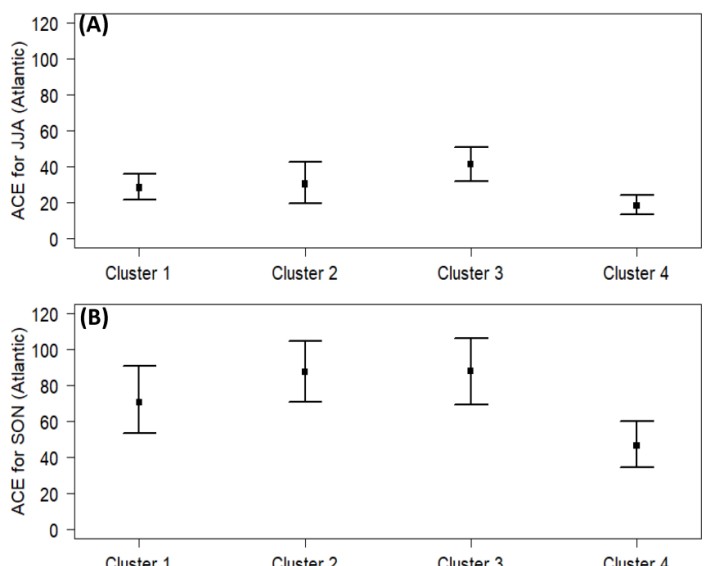

**Figure 7.** The 95% bootstrap confidence intervals of mean ACE for each cluster in the Atlantic basin for JJA (**A**) and SON (**B**). When the upper or lower confidence interval (outer bars) does not overlap the median (central point) of a given pairing, the pairing shows statistical significance.

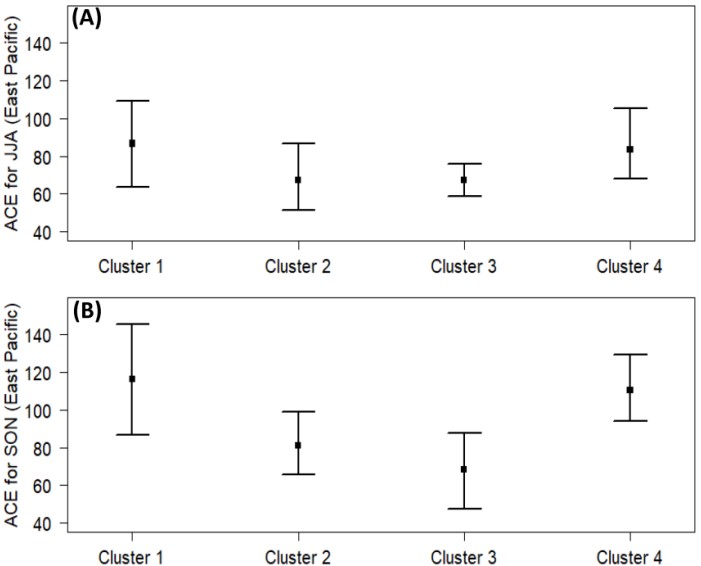

**Figure 8.** The 95% bootstrap confidence intervals of mean ACE for each cluster in the Eastern Pacific basin for JJA (**A**) and SON (**B**). When the upper or lower confidence interval (outer bars) does not overlap the median (central point) of a given pairing, the pairing shows statistical significance.

In the Atlantic basin, ACE during JJA is statistically significantly lower in cluster 4 than for all other clusters and is significantly higher in cluster 3 (Figure 7A). These findings indicate that the prevalence of EP El Niño tends to suppress TC activity in the Atlantic, while strong CP La Niña events greatly enhance TC activity. These results are in alignment with prior research on the role of ENSO phase in TC frequency [58]. They suggest that the impact of El Niño on the Walker Circulation results in increased vertical wind shear and decreased hurricane activity in the North Atlantic.

However, CP El Niño events, represented in cluster 1, exert a lesser influence compared to EP El Niño events in reducing TC activity in the Atlantic [59]. This difference may be attributed, in part, to the insufficient strength of these events in warming the tropical atmosphere, a crucial factor contributing to increased wind shear in the Atlantic basin [59]. Patricola et al. [60] found CP El Niño events to significantly hinder TC development in the Atlantic, but only compared to EP El Niño events of similar SST magnitudes. This is attributed to the lesser warming needed near the equatorial Central Pacific's warm pool to impact the Walker Circulation's strength, consequently amplifying vertical wind shear in the Atlantic basin. Thus, the smaller SSTAs in cluster 1 (CP El Niño) compared to cluster 4 (EP El Niño) may be the primary reason for the statistically significant decrease in TC activity associated with cluster 4.

La Niña events are associated with reduced wind shear in the Atlantic, creating favorable conditions for TC development [59]. Interestingly, cluster 3 (CP La Niña) has statistically significantly higher ACE than cluster 2 (EP La Niña), suggesting that CP La Niña events have a more pronounced influence on increasing ACE in the Atlantic than EP La Niña events. This is a novel result, as minimal research has explicitly delved into the influence of different La Niña flavors on TC activity in the Atlantic. Future research is needed to explore the physical mechanisms of La Niña flavors on TCs in the Atlantic basin.

For SON, cluster 4 again demonstrates a statistically significant decrease in ACE relative to all other clusters, while clusters 2 and 3 showed statistically significant increases in ACE relative to cluster 4 (Figure 7B). Expectedly, higher magnitudes of accumulated ACE in the later TC season (SON) express higher variability (noted by the spreads in the bootstrap confidence intervals in Figure 7B). The SON period tends to experience more and stronger TCs, resulting in larger ACE values compared to those observed during JJA. Additionally, the statistical distinction in EP and CP La Niña events (clusters 2 and 3) is lost when extending to SON, which suggests that CP La Niña events tend to result in

earlier accumulations of ACE during the TC season (a higher frequency of early-season TCs during CP La Niña years).

For Eastern Pacific ACE during JJA, the results show considerably higher variability than in the Atlantic, limiting the statistical significance of the JJA results (Figure 8). However, cluster 4 exhibits statistically significantly higher ACE compared to clusters 2 and 3, indicating the impact of EP El Niño conditions in contributing to higher TC activity compared to La Niña in clusters 2 and 3 [61] (Figure 8A). Unlike the Atlantic, CP and EP El Niño are statistically indistinguishable, though both show elevated ACE relative to La Niña conditions in the East Pacific. During SON, both clusters 1 and 4 (CP and EP El Niño) show statistically significant increases over clusters 2 and 3 (Figure 8B). Additionally, all clusters but cluster 3 show reduced ACE variability, suggesting that ACE predictability via these SST patterns may be increasing across the duration of the TC season. Cluster 1, characterized by weak El Niño conditions and a positive PMM, may synergistically enhance TC activity in the Pacific compared to clusters 2 and 3 [62].

## 4. Discussion and Conclusions

This paper utilizes hierarchical and k-means clustering techniques to identify established off-equatorial and ENSO-related SST patterns in the Eastern Pacific spanning from 1950 to 2021. A summary of the SSTA patterns, dominant climate drivers, and frequency trends is presented in Table 2. While previous research heavily relies on EOFs to understand tropical Pacific SST variability [11,22–24,30–33], this study highlights the use of cluster analysis as advantageous over EOFs for identifying SST patterns and understanding their climatic impacts. Cluster analysis directly constructs spatial patterns and events from observed SST fields without relying on variability fractions, and it does not assume linearity, allowing for the inclusion of nonlinear relationships [35,36]. This approach, rarely employed in this context, proves advantageous in identifying prevalent SST patterns aligned with ENSO flavors, offering greater utility in downstream climatic impact studies. This paper demonstrates this utility through an example application using ACE to identify statistically significant differences in ACE by the clusters' SST patterns and ENSO flavors.

Cluster 1 is characterized by a horseshoe pattern of positive anomalies in the subtropical North Pacific, Central Pacific, and central and eastern South Pacific. A strong influence from the positive PMM phase is observed during MAM, which is a precursor to CP El Niño in DJF for 38% of cases. The frequency of cluster 1 events increased during the study period, likely attributed to a general upward trend in positive PMM events. Cluster 2 is characterized by negative anomalies in the eastern equatorial Pacific that extend into the South Pacific and to the central equatorial Pacific in the winter. This cluster is predominately influenced by EP La Niña and easterly wind anomalies. The frequency of cluster 2 events exhibited an increasing trend during the study period. Cluster 3 exhibits negative anomalies in the eastern and Central Pacific, with the strongest cooling along the equator in the Central Pacific and within the subtropical regions of the Eastern Pacific. This cooling is likely due to a combination of La Niña, which is prevalent during fall and winter, and negative PMM events, which are prevalent for all three-month periods. A CP La Nina during DJF that follows a negative PMM during the spring is observed in 92% of the cases. The transition of a negative PMM during spring into a CP La Niña in the subsequent winter is supported by the strong presence of northeasterly wind anomalies across the majority of the subtropical North Pacific and equatorial Central Pacific, which are associated with the WES feedback. The decline of negative PMM events is likely the primary factor contributing to the decrease in this cluster over the study period. Cluster 4 is characterized by a large area of positive anomalies in the Eastern Pacific, with the strongest warming near the equator. This cluster is strongly influenced by EP El Niño with an absence of a strongly positive PMM during the spring. Wind anomalies are predominately easterly in the subtropical North Pacific, while strong westerly anomalies dominate the equatorial Pacific during the winter. Over the study period, cluster 4 events slightly decreased likely due to the slight decline in negative PMM events [54].

**Table 2.** Summary of the spatial patterns of the SSTAs, the dominant climate drivers, and the frequency trend from 1950 to 2019.

| Cluster | SSTA Pattern | Dominant Climate Drivers | Trend (1950–2019) |
|---|---|---|---|
| 1 | Anomalous warming in the subtropical North Pacific, Central Pacific, and central and eastern South Pacific. | Positive PMM and CP El Niño | Increased |
| 2 | Anomalous cooling in the eastern and central equatorial Pacific with small regions of anomalous cooling off the coast of Mexico. | EP La Niña | Increased |
| 3 | Anomalous cooling in the Eastern and Central Pacific, with the most intense cooling in the equatorial Central and subtropical Central and Eastern Pacific. | Negative PMM and CP La Niña | Decreased |
| 4 | Anomalous warming in the Central and Eastern Pacific, with significant warming near the equator in the Central and Eastern Pacific. | EP El Niño | Decreased |

This study also investigates connections between ACE in the Atlantic and Eastern Pacific basins and SST patterns using cluster member years. In the Atlantic basin throughout JJA, EP El Niño tends to suppress tropical cyclone (TC) activity, whereas CP La Niña events have an enhancing effect. La Niña occurrences in clusters 2 and 3 contribute to an increased ACE during SON, with CP La Niña in cluster 3 demonstrating a notably higher ACE compared to EP La Niña in cluster 2. This specific result needs further investigation into the underlying physical mechanisms governing the diverse impacts of various La Niña flavors on TCs in the Atlantic basin. In the Eastern Pacific during JJA, ACE exhibits greater variability, and cluster 4 exhibits a statistically significantly higher ACE than clusters 2 and 3, underscoring the influence of EP El Niño conditions on increased TC activity. During SON, both clusters 1 and 4 indicate statistically significant increases in ACE over clusters 2 and 3, indicating heightened TC activity. Both the weak El Niño event and positive PMM in cluster 1 could contribute to intensifying TC activity in the Eastern Pacific [62].

Overall, clusters 1–4 present an objective set of SSTA patterns that encompass both equatorial and off-equatorial regions in the eastern tropical Pacific found via unsupervised machine learning methods. However, this study has some limitations. One factor to consider is the relatively small sizes of certain clusters, which could influence the spatial distribution of the SSTAs. This issue can be resolved as a larger SST dataset becomes available over time since the most reliable record of SST data began in 1950. This time frame also limits the sample size of ENSO events, as Ren et al. [63] suggest there have only been 19 El Niño events and 14 La Niña events since 1950. This limitation, as with the SST data, can be overcome with additional years of data.

While some limitations exist, this study utilizes a multi-decadal SSTA dataset to identify dominant observational structures that can be directly linked to individual years and downstream climate factors such as TC ACE. We show that the application of these patterns for different downstream effect studies (e.g., precipitation studies, severe weather studies) can provide new insights into the links between Pacific SSTAs and these phenomena. The methods used in this study provide more streamlined analysis tools for identifying the most prevalent SSTA patterns and utilizing these patterns to diagnose subsequent climatic impacts.

**Author Contributions:** Conceptualization, B.F., C.F. and A.M.; formal analysis, J.F., B.F. and A.M.; investigation, A.M.; methodology, J.F., B.F., C.F. and A.M.; supervision, B.F., C.F., A.M. and J.R.; validation, J.F.; visualization, J.R.; writing—original draft, J.F., C.F. and A.M.; writing—review and editing, J.F., B.F., C.F., A.M. and J.R. All authors have read and agreed to the published version of the manuscript.

**Funding:** This research received no external funding.

**Data Availability Statement:** The NOAA Extended Reconstructed Sea Surface Temperature (ERSST) Version 5 data are openly available from NCEI's Extended Reconstructed SST site: https://www.ncei. noaa.gov/products/extended-reconstructed-sst, accessed on 22 March 2022. The ECMWF ERA5 Reanalysis data are openly available from the Copernicus site: https://cds.climate.copernicus.eu/ cdsapp#!/dataset/reanalysis-era5-single-levels?tab=overview, accessed on 2 January 2024. The Oceanic Niño Index (ONI) data are freely available on the CPC's site: https://www.cpc.ncep.noaa. gov/products/analysis_monitoring/ensostuff/ONI_change.shtml, accessed on 22 March 2022. The Pacific Meridional Mode (PMM) data are openly available on the University of Wisconsin's Meridional Mode site: https://www.aos.wisc.edu/~dvimont/MModes/Home.html, accessed on 22 March 2022. Accumulated Cyclone Energy data are openly available through NOAA's Physical Science Laboratory site: https://psl.noaa.gov/gcos_wgsp/Timeseries/Hurricane/, accessed on 3 February 2023.

**Acknowledgments:** The first author would like to thank the contributing authors for their skillful guidance and expertise in improving and completing this paper. Their contributions are greatly appreciated. An additional thanks to Kimberly Wood for her expertise in tropical cyclones and detailed editing of previous manuscripts.

**Conflicts of Interest:** The authors declare no conflicts of interest.

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
