# Peer review of "Quantifying Downstream Climate Impacts of Sea Surface Temperature Patterns in the Eastern Tropical Pacific Using Clustering"

_climate, doi:10.3390/cli12050071_

Round 1
Reviewer 1 Report
Comments and Suggestions for Authors
Both ENSO (El Nino/La Nina) and Pacific meridional mode are hot topics which have been widely explored. Due to their important influences on climate and weather in other regions, they will continue to be investigated in the future. But their close relationship also poses a big challenge on how to efficiently separate one event with another. Authors of this manuscript propose a novel method, a cluster analysis with machine learning which can effectively separate them. This manuscript has been prepared well from abstract to the final summary and conclusion. However, some small but important issues should be paid on attentions. I list them here as reference for the authors.
1. Pacific meridional mode is a key factor in this manuscript, so the definition of its index should be explained in detail. Although PMM index that the authors used is from the University of Wisconsin’s group, the general information for its methodology should be provided in this paper and please provide associated references as well.
2. So does for accumulative cyclone energy (ACE). The authors should generally explain how to calculate ACE and provide the information for where the geographic regions of ACE in the Atlantic Ocean and the eastern Pacific Ocean locate.
3. Lines 49-52, impacts of El Nino/La Nina on wintertime atmospheric circulation may not be only through Walker circulation, and its influences on middle and high latitudes could be through teleconnecting and changing Hadley circulation.
4. Lines 161-167, can the authors explain more on how to remove ENSO variability from PMM index?
5. Line 281 and Fig.3, the linear increase of no. of La Nina events in the whole period (1950-2019) is small, and a striking feature showing Fig.3 is its increase in the first half period (1950-1989) but decrease in the second half period (1980-2019). The authors should mention this in the paper. I am also curious if this phenomenon is consistent with ENSO index? The same issue is for no. of El Ninos showing in Fig.5E because I concern if the cluster analysis is a effective method to separate ENSO from PMM.
6. From Fig.2-5, the authors find that positive PMM (cluster 1) may generate CP El Nino and negative PMM closely relate to CP La Nina (cluster 3). However, I can’t get those from Table 1 and the scatterplots Fig.6a and 6c. Can the authors explain more about the reasons?
7. As explanations for impacts of SSTA on ACE in the eastern Pacific and Atlantic Ocean, is there a similar change on ACE in the western Pacific Ocean where have more tropical cyclones annually and cause more damages on economy and human lives?
Reviewer 2 Report
Comments and Suggestions for Authors
This paper clustered the SSTA pattern in the Eastern tropical Pacific. The 4 clusters are related different ENSO flavors and are associated with ACE changes in the Atlantic and Pacific. The results are reasonable. Some modifications are still needed before publication.
1. Delete “unsupervised learning” in the title. Just using “clustering” or “cluster” instead.
2. A time series of monthly ONI (with 4 colors indicating 4 clusters) can be helpful for visual display about the interannual difference and their relationship among the 4 clusters.
3. Some related paper about ENSO cluster should be cited.
Su, J., Lian, T., Zhang, R., & Chen, D. (2018). Monitoring the pendulum between El Niño and La Niña events. Environmental Research Letters, 13(7), 074001.
Zhao, J., Kug, J. S., Park, J. H., & An, S. I. (2020). Diversity of North Pacific meridional mode and its distinct impacts on El Niño‐southern oscillation. Geophysical Research Letters, 47(19), e2020GL088993.
